# TNF-α exacerbates SARS-CoV-2 infection by stimulating CXCL1 production from macrophages

**Moe Kobayashi[1], Nene Kobayashi[1], Kyoka Deguchi[1], Seira Omori[1], Minami Nagai[1], Ryutaro Fukui[2], Isaiah Song[3], Shinji Fukuda[3,4,5,6], Kensuke Miyake[2], Takeshi Ichinohe**[1] *

**1** Division of Viral Infection, Department of Infectious Disease Control, International Research Center for Infectious Diseases, Institute of Medical Science, The University of Tokyo, Tokyo, Japan, **2** Division of Innate Immunity, Department of Microbiology and Immunology, The Institute of Medical Science, The University of Tokyo, Tokyo, Japan, **3** Institute for Advanced Biosciences, Keio University, Mizukami, Kakuganji, Tsuruoka, Yamagata, Japan, **4** Gut Environmental Design Group, Kanagawa Institute of Industrial Science and Technology,Tonomachi, Kawasaki, Kanagawa, Japan, **5** Transborder Medical Research Center, University of Tsukuba, Tennodai, Tsukuba, Ibaraki, Japan, **6** Laboratory for Regenerative Microbiology, Juntendo University Graduate School of Medicine, Hongo, Bunkyo-ku, Tokyo, Japan

* ichinohe@ims.u-tokyo.ac.jp

**Data Availability Statement:** All relevant data are within the manuscript and its Supporting Information files.

## Abstract

Since most genetically modified mice are C57BL/6 background, a mouse-adapted SARS-CoV-2 that causes lethal infection in young C57BL/6 mice is useful for studying innate immune protection against SARS-CoV-2 infection. Here, we established two mouse-adapted SARS-CoV-2, ancestral and Delta variants, by serial passaging 80 times in C57BL/6 mice. Although young C57BL/6 mice were resistant to infection with the mouse-adapted ancestral SARS-CoV-2, the mouse-adapted SARS-CoV-2 Delta variant caused lethal infection in young C57BL/6 mice. In contrast, MyD88 and IFNAR1 KO mice exhibited resistance to lethal infection with the mouse-adapted SARS-CoV-2 Delta variant. Treatment with recombinant IFN-α/β at the time of infection protected mice from lethal infection with the mouse-adapted SARS-CoV-2 Delta variant, but intranasal administration of recombinant IFN-α/β at 2 days post infection exacerbated the disease severity following the mouse-adapted ancestral SARS-CoV-2 infection. Moreover, we showed that TNF-α amplified by type I IFN signals exacerbated the SARS-CoV-2 infection by stimulating CXCL1 production from macrophages and neutrophil recruitment into the lung tissue. Finally, we showed that intravenous administration to mice or hamsters with TNF protease inhibitor 2 alleviated the severity of SARS-CoV-2 and influenza virus infection. Our results uncover an unexpected mechanism by which type I interferon-mediated TNF-α signaling exacerbates the disease severity and will aid in the development of novel therapeutic strategies to treat respiratory virus infection and associated diseases such as influenza and COVID-19.

## Author summary

Coronavirus disease 2019 (COVID-19) cause severe morbidity and mortality worldwide. Although mounting evidence indicates that the virus-induced cytokine storm associates

**Funding:** This work was supported in part by research grants from the Japan Agency for Medical Research and Development (AMED) (JP233fa627001 to T.I. and K.M, JP22gm1010009 to S.F.), JSPS KAKENHI (22H03541 to S.F.), JST ERATO (JPMJER1902 to S.F.), the Food Science Institute Foundation (to S.F.). The funders had no role in study design, data collection and analysis, decision to publish, or preparation of the manuscript.

**Competing interests:** The authors have declared that no competing interests exist.

with severity of COVID-19, the pathological role of inflammatory cytokines in severe COVID-19 remains unknown. Here, we demonstrated that the virus-induced pulmonary TNF-α amplified by MyD88 and type I interferon receptor signals exacerbated the SARS-CoV-2 infection by stimulating CXCL1 production from macrophages. In addition, we found that TNF-α protease inhibitor 2 alleviated the severity of SARS-CoV-2 and influenza virus infection in mice and hamsters. Our results provide important insights into the role of type I interferon-mediated TNF-α signaling in neutrophil-induced lung pathology and will aid the development of novel therapeutic strategies to treat respiratory viral infections and associated diseases.

## Introduction

While human ACE2 (hACE2) transgenic mice are widely used worldwide in SARS-CoV-2 research, studying innate immune signals required for protection against SARS-CoV-2 infection *in vivo* requires crossing hACE2 transgenic mice with innate immune-related gene-deficient mice [1]. A mouse model of SARS-CoV-2 based on adeno-associated virus (AAV)-mediated expression of hACE2 in respiratory tract is useful model for COVID-19 pathogenesis and protection [2]. Similarly, a mouse-adapted SARS-CoV-2 is convenient for studying innate immune signals required for protection against SARS-CoV-2 infection *in vivo*. However, previously established mouse-adapted SARS-CoV-2 causes lethal infection only in aged Balb/c or C57BL/6 mice but not 6-week-old young C57BL/6 mice [3–6]. Since most genetically modified mice are C57BL/6 background, a mouse-adapted SARS-CoV-2 that causes lethal infection in young C57BL/6 mice could be more powerful tool for studying innate immune signals required for protection against SARS-CoV-2 infection.

A previous study using a mouse model of SARS-CoV-2 based on AAV-mediated expression of hACE2 demonstrated that type I interferon (IFN) signaling do not control SARS-CoV-2 replication *in vivo* but are significant drivers of pathological responses [2]. In contrast, Ogger et al. showed that type I IFN signaling is essential for suppressing SARS-CoV-2 replication and inflammatory myeloid cell recruitment to the lung of AAV-hACE2-transduced mice following SARS-CoV-2 infection [7]. In addition, a recent study demonstrated that the STING-deficient K18-hACE2 mice show no difference in weight change and survival following SARS-CoV-2 infection [1]. However, the role of other innate immune signals in severity of SARS-CoV-2 infection *in vivo* remains unknown.

The objective of this study is to establish a lethal SARS-CoV-2 infection model in 6-week-old mice, which serves as a critical platform for testing novel therapeutics and vaccines, as well as to elucidate the role of innate immune signals in the severity of COVID-19. Unlike models in 10–12-week-old C57BL/6J mice [3,5], which may still be influenced by age-related factors, the use of younger mice minimizes potential confounding effects while providing a standardized and reproducible system. Additionally, this model allows for the investigation of severe disease mechanisms relevant to younger populations, including rare but significant cases of severe COVID-19 in pediatric and adolescent patients. Therefore, we attempted to establish a mouse-adapted SARS-CoV-2 strain that causes lethal infection in young mice. In this study, we establish mouse-adapted ancestral SARS-CoV-2 and the Delta variants by serial passaging 80 times in C57BL/6 mice (named ancestral P80 and Delta P80 viruses, respectively). We demonstrate that MyD88 and type I IFN signaling exacerbates the Delta P80 virus infection. In addition, type I IFN signaling amplifies secretion of pulmonary inflammatory cytokines including tumor necrosis factor alpha (TNF-α), which in turn stimulates CXCL1 production

from macrophages and stimulates neutrophil recruitment into the lung tissue. Further, we find that intravenous administration to mice or hamsters with TNF-α protease inhibitor 2 alleviated the severity of SARS-CoV-2 and influenza virus infection.

## Results

### Generation of a mouse-adapted ancestral SARS-CoV-2

Leist et al. demonstrated that 1-year-old mice were highly susceptible to mouse-adapted SARS-CoV-2 [4]. Thus, we first infected 83-weeks-old aged C57BL/6 mice intranasally with an ancestral SARS-CoV-2 (SARS-CoV-2/UT-NCGM02/human/2020) to generate a mouse-adapted virus [8]. Then, we collected the lung washes at 3 days post infection (p.i.) and inoculated them into VeroE6/TMPRSS2 cells to propagate the SARS-CoV-2 (refer to as P1) (Fig 1A). We confirmed that these viruses were gradually adapted in aged C57BL/6 mice during 5 serial passages by reverse transcription-quantitative polymerase chain reaction (RT-qPCR) (Fig 1B). After 5 serial passages in aged mice, we next infected 6-weeks-old young C57BL/6 mice intranasally with the P5 virus (Fig 1A). Then, we collected the lung washes at 3 days p.i. and inoculated them into VeroE6/TMPRSS2 cells to propagate the P6 virus (Fig 1A). Although the levels of SARS-CoV-2 nucleoprotein RNA in the lung washes at 3 days p.i. were gradually reduced during 5 serial passages in young mice, these viruses were gradually adapted in young mice during additional 3 passages (Fig 1B). After 10 serial passages in young mice, we serially passaged the lung washes of the virus-infected young mice every 3 days without virus propagation in VeroE6/TMPRSS2 cells to generate the ancestral P80 virus (Fig 1A). The ancestral P80 viral stocks used in the experiments were propagated in VeroE6/TMPRSS2 cells. Next-generation sequencing analysis revealed that the ancestral P80 virus contained 27 amino acid substitutions that were distributed within the ORF1ab, S, 3a, E, M, 7a, N and 10 genes, respectively (Fig 1C).

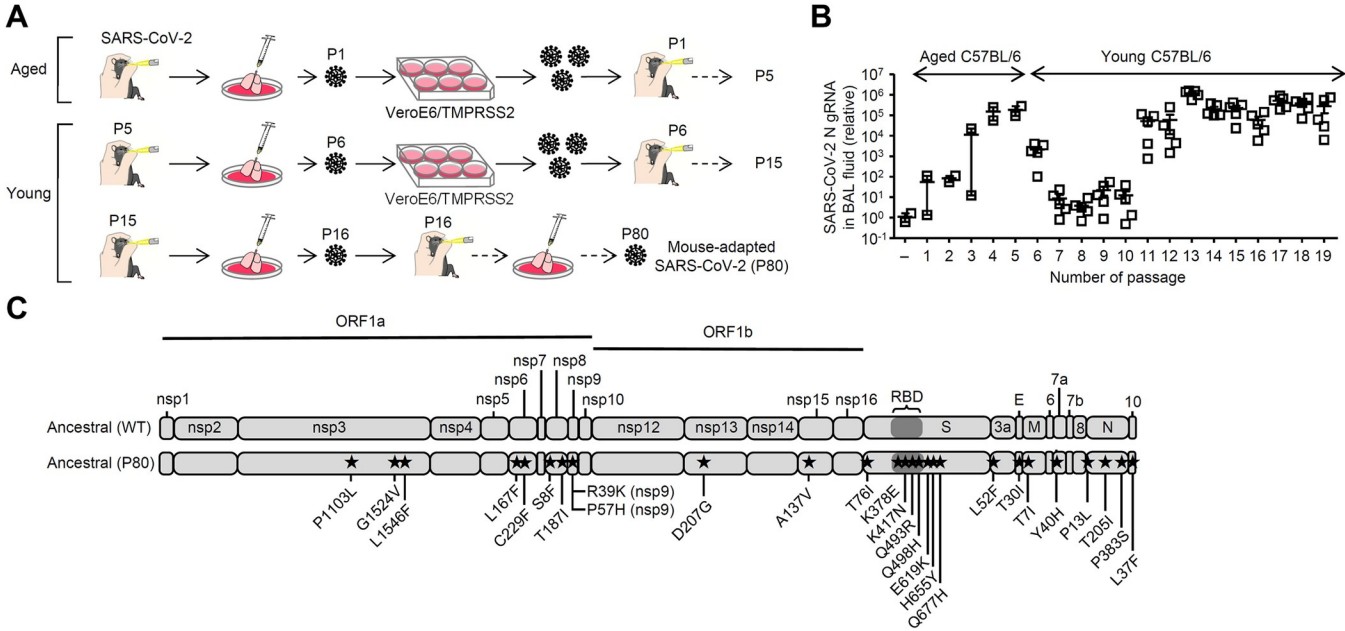

**Fig 1. Generation of a mouse-adapted ancestral SARS-CoV-2. (A)** Schematic representation of experimental setup. **(B)** Total RNAs were extracted from lung washes at 3 days p.i. and SARS-CoV-2 N gRNA levels were assessed by quantitative reverse transcription PCR. **(C)** Schematic diagram of SARS-CoV-2 genome and all the adaptive mutations identified in the ancestral P80 virus. Nonsynonymous mutations were compared to the original ancestral SARS-CoV-2 (WT). Each symbol indicates individual values (B). Data are mean ± s.e.m. (B).

## Pathogenesis of a mouse-adapted ancestral SARS-CoV-2 in laboratory mice

We next examined pathogenesis of mouse-adapted the ancestral P80 virus in standard laboratory mice. After the ancestral P80 virus infection, young Balb/c and C3H mice succumbed to disease by 5 days p.i. (Fig 2A–2D). In contrast, young C57BL/6 mice exhibited ~25% weight loss and recovered by 10 days p.i. (Fig 2E and 2F). The virus titers were significantly elevated in the lung of young Balb/c mice compared with those of young C57BL/6 mice (Fig 2G).

## The Delta P80 virus causes lethal infection in young C57BL/6 mice

Since the ancestral P80 virus did not cause lethal infection in young C57BL/6 mice, we next tried to generate a mouse-adapted SARS-CoV-2 Delta variant (hCoV-19/Japan/TY11-927-P1/2021) [9]. As described in Fig 1A, we first passed a SARS-CoV-2 Delta variant five times in aged mice and then additional 10 times in young mice. After the Delta P18 or P29 virus infection, young C57BL/6 mice did not reduce their body weight (Fig 3A and 3B). In addition, young C57BL/6 mice infected with the Delta P41 virus exhibited ~10% weight loss and recovered by 6 days p.i. (Fig 3C). In contrast, the Delta P61 and P80 viruses caused lethal infection in young C57BL/6 mice (Fig 3D–3F). The Delta P80 virus was found to exhibit enhanced replication in the lung of young C57BL/6 mice compared with the mouse-adapted Delta P18, P29, P41, or P61 variants without affecting viral replication in VeroE6/TMPRSS2 cells (Fig 3G and 3H). Next-generation sequencing analysis revealed that the Delta P80 virus contained 27 amino acid substitutions that were distributed within the ORF1ab, S, E, M and N genes, respectively (Fig 3I).

So far, we established two mouse-adapted SARS-CoV-2 variants by serial passaging 80 times in C57BL/6 mice. Next, we compared pathogenesis of the ancestral and Delta P80 viruses in young C57BL/6 mice. While young C57BL/6 mice were resistant to the ancestral P80 virus infection, the Delta P80 virus caused lethal infection (Fig 4A and 4B). In addition, the Delta P80 virus caused severe pulmonary edema in young C57BL/6 mice, while the ancestral P80 virus caused partial lung inflammation (Fig 4C and 4D). The virus titers, virus-induced

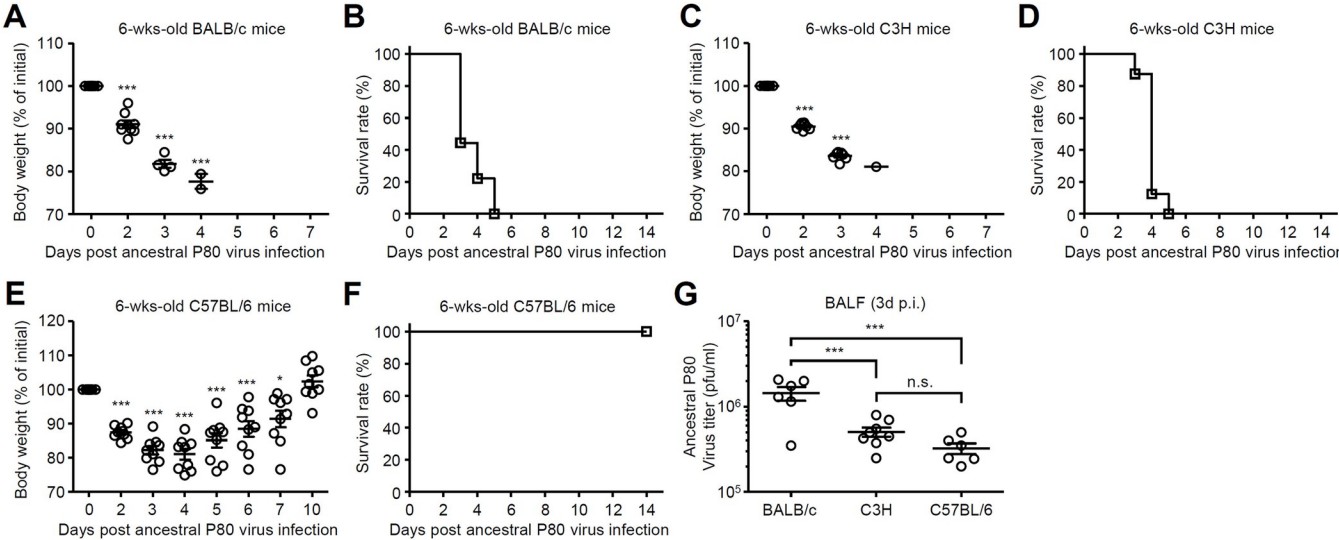

**Fig 2. Pathogenesis of a mouse-adapted ancestral SARS-CoV-2 in laboratory mice. (A-F)** BALB/c, C3H, and C57BL/6 mice were infected intranasally with $1 \times 10^5$ pfu of the ancestral P80 virus. Weight loss (A, C and E) and mortality (B, D and F) were monitored for 14 days. **(G)** The lung washes were collected at 3 days p.i. and viral titers were determined by standard plaque assay. Each symbol indicates individual values (A, C, E and G). Data are mean ± s.e.m. (A, C, E and G). Statistical significance was analyzed by two-way analysis of variance (ANOVA) (A, C, E and G). *$P < 0.05$, ***$P < 0.001$, n.s., not significant.

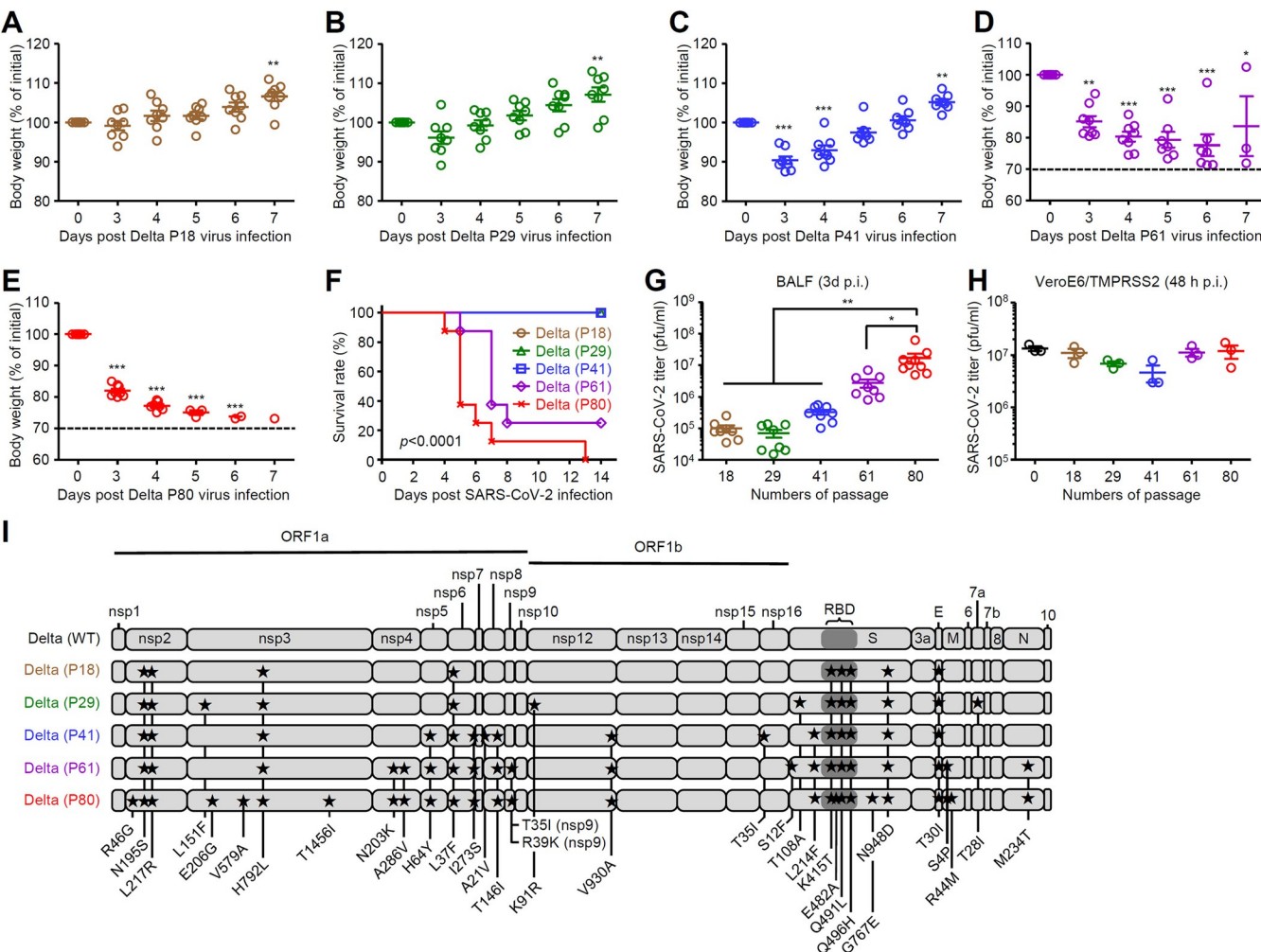

**Fig 3. The Delta P80 virus causes lethal infection in young C57BL/6 mice. (A-G)** Six-week-old C57BL/6 mice were infected intranasally with $1 \times 10^5$ pfu of the Delta P18 (A, F and G), P29 (B, F and G), P41 (C, F and G), P61 (D, F and G), or P80 virus (E, F and G). Weight loss (A-E) and mortality (F) were monitored for 14 days. The dashed line indicates the limit of endpoint (D and E). The lung washes were collected at 3 days p.i. and viral titers were determined by standard plaque assay (G). **(H)** VeroE6/TMPRSS2 cells were infected with the original SARS-CoV-2 Delta variant, mouse-adapted Delta P18, P29, P41, P61, or P80 virus. Cell-free supernatants were collected at 48 h p.i. and analyzed for virus titer by standard plaque assay using VeroE6/TMPRSS2 cells. **(I)** Schematic diagram of SARS-CoV-2 genome and all the adaptive mutations identified in the Delta P18, P29, P41, P61, and P80 virus. Nonsynonymous mutations were compared to the original SARS-CoV-2 Delta variant (WT). Each symbol indicates individual values (A-E, G and H). Statistical significance was analyzed by two-way analysis of variance (ANOVA) (A-E and G), or two-sided log-rank (Mantel-Cox) test (F). *P < 0.05, **P < 0.01, ***P < 0.001.

proinflammatory cytokines, and neutrophil recruitment were significantly elevated in the lung of the Delta P80 virus-infected mice compared with those of ancestral P80 virus infected mice (Figs 4E–4J and S1). While the primary target of these two variants was the CD45.2 negative epithelial cells in the lung (Figs 4K and S2), the Delta P80 virus also infected a large number of alveolar macrophages and dendritic cells in the lung tissue (S3–S7 Figs). Next, we examined the possibility that the Delta P80 virus infects tissues other than the lung in mice. However, we were unable to detect viral RNA in the brain, heart, liver or kidney of mice infected with the ancestral or Delta P80 viruses (Fig 4L and 4M). These results suggest that the differences in pathogenicity between these two variants in young C57BL/6 mice are likely due to differences in viral replication or virus-induced inflammatory responses in the lung, and that the Delta variant has not acquired the ability to infect other organs outside the lung.

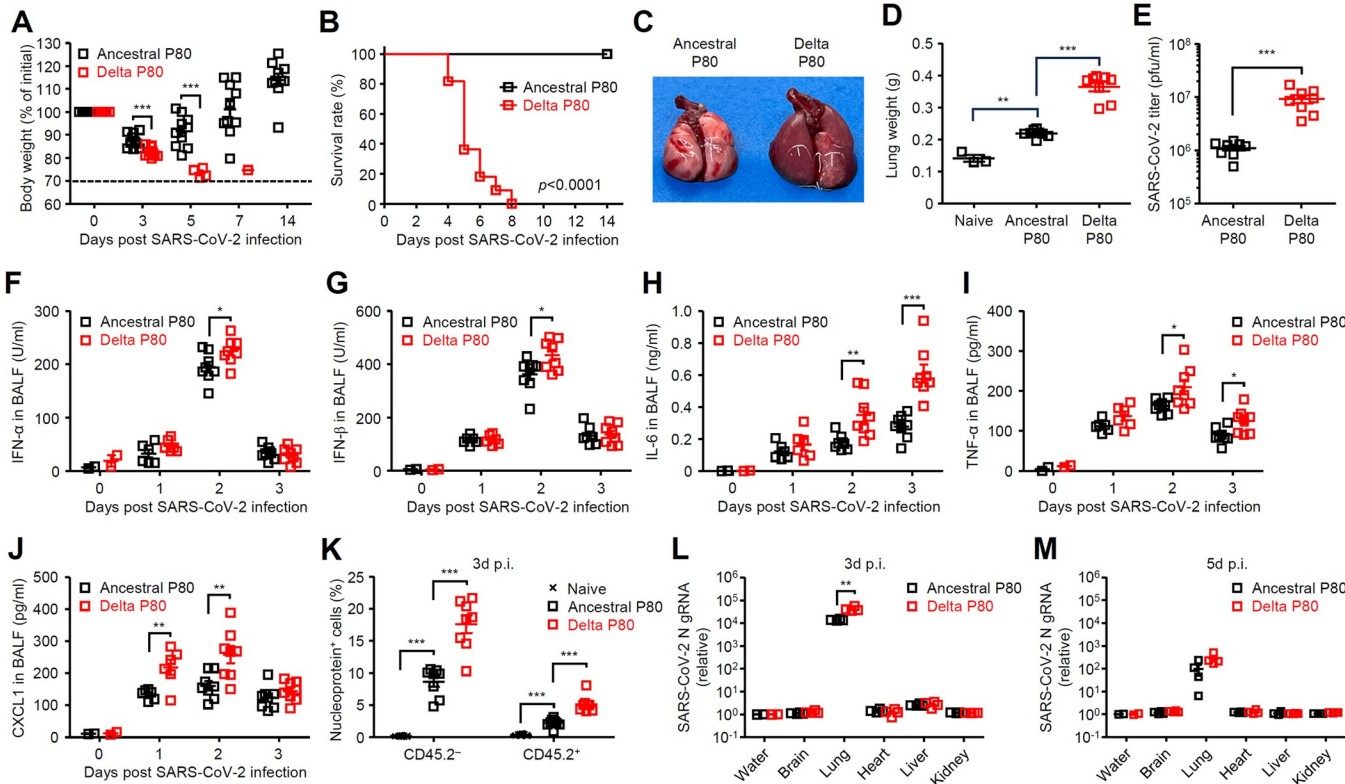

**Fig 4. The Delta P80 virus causes severe pneumonia in young C57BL/6 mice.** Six-week-old C57BL/6 mice were infected intranasally with $1 \times 10^5$ pfu of the ancestral or Delta P80 virus. **(A and B)** Weight loss (A) and mortality (B) were monitored for 14 days. The dashed line indicates the limit of endpoint (A). **(C and D)** Gross lung pathology (C) and total lung weight (D) of mice infected with the ancestral or Delta P80 viruses at 4 days p.i.. **(E)** The lung washes were collected at 3 days p.i. and viral titers were determined by standard plaque assay. **(F-J)** The lung washes were collected at indicated time points and analyzed for IFN-$\alpha$ (F), IFN-$\beta$ (G), IL-6 (H), TNF-$\alpha$ (I), and CXCL1 (J) by ELISA. **(K)** The lung was collected from the virus-infected mice at 3 days p.i.. The single-cell-suspensions of lung samples were stained with anti-SARS-CoV-2 nucleoprotein and anti-CD45.2 antibodies. The ratio of the nucleoprotein-positive cells among CD45.2-positive or CD45.2-negative cells are shown. **(L and M)** Total RNAs were extracted from indicated tissues at 3 (L) and 5 (M) days p.i. The levels of SARS-CoV-2 N gRNA were assessed by quantitative reverse transcription PCR. Each symbol indicates individual values (A, D-M). Statistical significance was analyzed by two-tailed unpaired Student's $t$ test (A, E-J, L, and M), two-sided log-rank (Mantel-Cox) test (B), or two-way analysis of variance (ANOVA) (D and K). $^*P < 0.05$, $^{**}P < 0.01$, $^{***}P < 0.001$.

## MyD88 and IFNAR1 signals exacerbate SARS-CoV-2 infection

To examine innate immune signals required for protection against SARS-CoV-2 infection, we took advantage of the ancestral and Delta P80 viruses with different pathogenicity in young C57BL/6 mice. We first infected wild-type (WT) and MyD88 KO mice with the ancestral P80 virus. However, both WT and MyD88 KO mice recovered from ancestral P80 virus infection (Fig 5A and 5B). Similarly, no significant differences were observed in body weight change or survival rate between MyD88-deficient and WT mice following a sublethal dose ($1 \times 10^4$ pfu) of the Delta P80 virus infection (S8 Fig). Next, we infected WT and MyD88 KO mice with a lethal dose ($1 \times 10^5$ pfu) of the Delta P80 virus. Interestingly, MyD88 KO mice exhibited resistance to lethal infection with the Delta P80 virus infection (Fig 5C and 5D). Similarly, IFNAR1 KO mice were highly resistant to the Delta P80 virus infection (Fig 5E and 5F). Although the virus titers were comparable between WT and IFNAR1 or MyD88 KO mice (Fig 5G and 5H), the levels of IFN-$\alpha$, IFN-$\beta$, IFN-$\lambda$, and TNF-$\alpha$, but not IL-6 were significantly reduced in the lungs of MyD88 and IFNAR1 KO mice compared with those of WT mice at 2 days p.i. (Fig 5I–5M). In contrast to these proinflammatory cytokines, the Delta P80 virus did not induce detectable

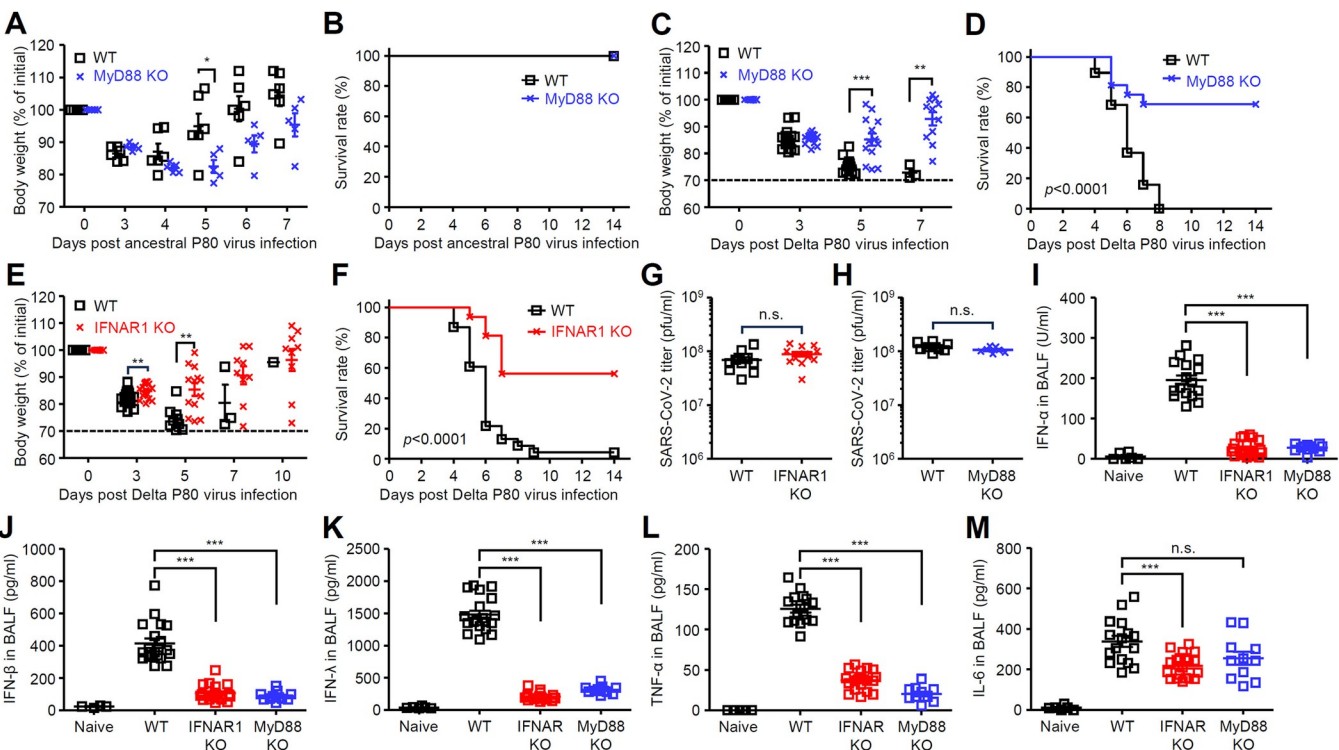

**Fig 5. MyD88 and IFNAR1 KO mice are resistant to Delta P80 virus infection. (A and B)** Six-week-old C57BL/6 WT or MyD88 mice were infected intranasally with 1×10⁵ pfu of the ancestral P80 virus. Weight loss (A) and mortality (B) were monitored for 14 days. **(C-M)** Six-week-old C57BL/6 WT, MyD88, or IFNAR1 KO mice were infected intranasally with 1×10⁵ pfu of the Delta P80 virus. Weight loss (C and E) and mortality (D and F) were monitored for 14 days. The dashed line indicates the limit of endpoint (C and E). The lung washes were collected at 3 days p.i. and viral titers were determined by standard plaque assay (G and H). The lung washes were collected at 2 days p.i. and analyzed for IFN-α (I), IFN-β (J), IFN-λ (K), TNF-α (L), and IL-6 (M) by ELISA. Each symbol indicates individual values (A, C, E and G-M). Statistical significance was analyzed by two-tailed unpaired Student's *t* test (A, C, E, G and H), two-sided log-rank (Mantel-Cox) test (B, D and F), or two-way analysis of variance (ANOVA) (I-M). *$P < 0.05$, **$P < 0.01$, ***$P < 0.001$, n.s., not significant.

levels of IL-1β in the lungs of WT, MyD88, or IFNAR1 KO mice at 2 days p.i. (S9 Fig). These data suggest that the virus-induced proinflammatory cytokines rather than virus burden in the lung may contribute lethality in the Delta P80 virus-infected mice.

Thus far, our data indicated that the virus-induced proinflammatory cytokines amplified by type I IFN signals exacerbate SARS-CoV-2 infection. Thus, we next examined the role of type I IFNs in disease severity of SARS-CoV-2 infection. Intranasal administration to mice with recombinant IFN-α and IFN-β (IFN-α/β) at the time of infection completely protected mice from the ancestral or Delta P80 virus infection, highlighting the importance of type I IFNs in defense against SARS-CoV-2 infection (Fig 6A–6C). In contrast, intranasal administration to mice with recombinant IFN-α/β at later time points (2 to 4 days p.i.) exacerbated the ancestral P80 virus infection in WT but not IFNAR1 KO mice (Fig 6D–6G). Further, MyD88 KO mice, which were resistant to the Delta P80 virus infection (Fig 5C and 5D), exacerbated the disease after intranasal administration of IFN-α/β at 2 days post infection (Fig 6H).

## TNF-α-CXCL1 axis exacerbates SARS-CoV-2 infection

Next, we wished to determine the mechanism by which type I IFN signals following SARS-CoV-2 infection exacerbate severity of the disease. Previous studies have indicated the detrimental role of neutrophils in severe COVID-19 [10–14]. Because the levels of IFN-α, IFN-β, IFN-λ, and TNF-α were significantly elevated in the lungs of the Delta P80 virus-infected WT

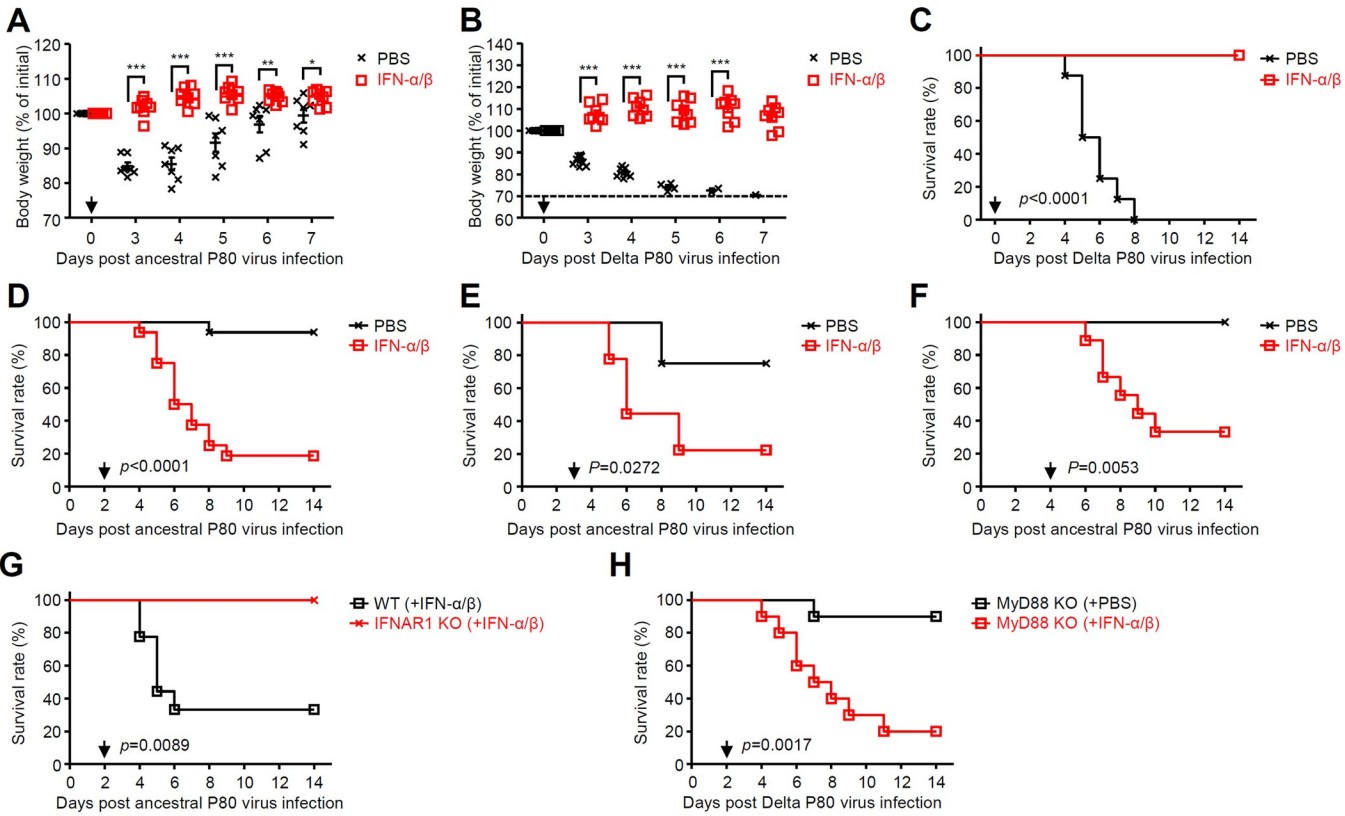

**Fig 6. Protective and detrimental roles of type I IFNs in SARS-CoV-2 infection.** (A-C) Six-week-old C57BL/6 mice were administered intranasally with $1\times10^5$ pfu of the ancestral or Delta P80 virus together with PBS or recombinant mouse IFN-$\alpha$ (1,250 unit) and IFN-$\beta$ (1.25 ng) (arrow). Weight loss (A and B) and mortality (C) were monitored for 14 days. The dashed line indicates the limit of endpoint (B). (D-H) Six-week-old C57BL/6 WT (D-G), IFNAR1 (G), or MyD88 KO (H) mice infected with $1\times10^5$ pfu of the ancestral (D-G) or Delta P80 virus (H) were administered intranasally with PBS or recombinant mouse IFN-$\alpha$ (1,250 unit) and IFN-$\beta$ (1.25 ng) at 2 (D, G and H), 3 (E), or 4 (F) days p.i. (arrow). Mortality was monitored for 14 days. Each symbol indicates individual values (A and B). Statistical significance was analyzed by two-tailed unpaired Student's *t* test (A and B), or two-sided log-rank (Mantel-Cox) test (C-H). *$P < 0.05$, **$P < 0.01$, ***$P < 0.001$.

mice compared with those of MyD88 and IFNAR1 KO mice (Fig 5I–5L), we first examined whether these cytokines are involved in production of CXCL1, which is one of the major che-moattractant of neutrophils, from macrophages. Although IL-1$\beta$ is an important mediator of CXCL1 production [15,16], it is unknown whether other cytokines stimulate CXCL1 produc-tion. Interestingly, treatment of bone marrow-derived macrophages with TNF-$\alpha$ significantly stimulated CXCL1 production in a dose-dependent manner (Fig 7A). We next examined the role of CXCL1 in disease severity of SARS-CoV-2 infection. Indeed, intranasal administration of CXCL1 at later but not early time points exacerbated the ancestral P80 virus infection (Fig 7B–7D). Following the Delta P80 virus infection, the levels of CXCL1 in the lung washes became apparent around day 2 and 3 p.i. (Fig 7E). In contrast, the levels of pulmonary CXCL1 were significantly suppressed in the lung washes of MyD88 or IFNAR1 KO mice compared with WT mice at 2 days p.i. (Fig 7F and 7G). We next examined whether the CXCL1-mediated exacerbation of SARS-CoV-2 infection can be reproduced by intranasal administration of TNF-$\alpha$. Intranasal administration of TNF-$\alpha$ at 2 days p.i. exacerbated the ancestral P80 virus infection by stimulating CXCL1 production and neutrophil recruitment into the lung tissue (Fig 7H–7J). In addition, MyD88 KO mice, which were resistant to the Delta P80 virus infec-tion, exacerbated the disease after intranasal administration of TNF-$\alpha$ at 2 days p.i. (S10 Fig).

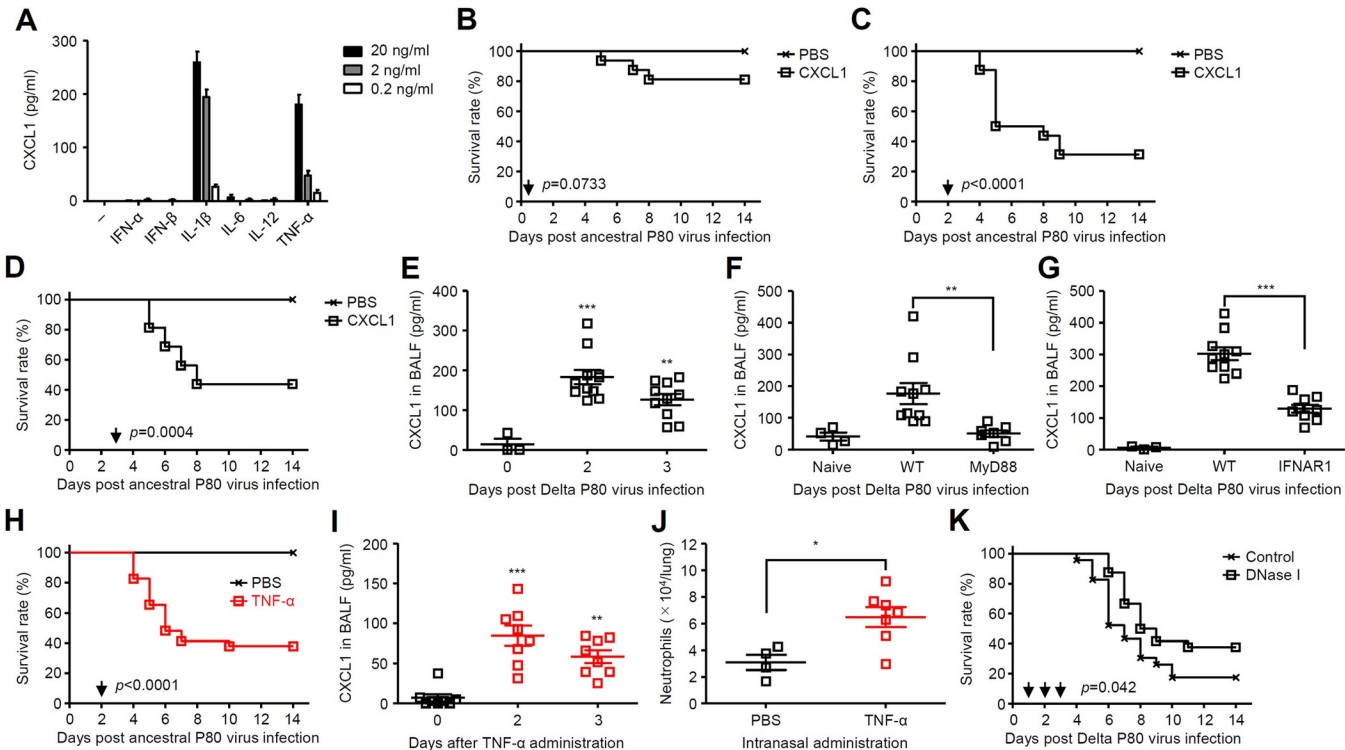

**Fig 7. TNF-α-CXCL1 axis exacerbates ancestral P80 virus infection. (A)** Bone marrow-derived macrophages were stimulated with indicated recombinant mouse cytokines. Cell-free supernatants were collected at 24 h p.i. and analyzed for CXCL1 by ELISA. (**B**-**D**) Six-week-old C57BL/6 mice infected with $1\times10^5$ pfu of the ancestral P80 virus were administered intranasally with PBS or recombinant mouse CXCL1 (2.5 μg) at 6 hours (B), 2 (C), or 3 (D) days p.i. (allow). Mortality was monitored for 14 days. **(E-G)** Six-week-old C57BL/6 WT (E-G), MyD88 (F), or IFNAR1 KO (G) mice were infected intranasally with $1\times10^5$ pfu of the Delta P80 virus. The lung washes were collected at 2 (E-G) or 3 (E) days p.i. and analyzed for CXCL1 by ELISA. **(H)** Six-week-old C57BL/6 WT mice infected with the ancestral P80 virus were administered intranasally with PBS or recombinant mouse TNF-α (2.5 μg) at 2 days p.i. (arrow). Mortality was monitored for 14 days. **(I and J)** Six-week-old C57BL/6 WT mice were administered intranasally with a recombinant mouse TNF-α (2.5 μg). The lung washes were collected at indicated time points and analyzed for CXCL1by ELISA (I). Three days later, leukocytes were isolated from the lung. The number of Ly6C+ Ly6G+ neutrophils were analyzed by flow cytometry (J). **(K)** Six-week-old C57BL/6 WT mice infected with the Delta P80 virus were administered intraperitoneally with a recombinant DNase I at indicated time points (arrow). Mortality was monitored for 14 days. Each symbol indicates individual values (E-G, I, and J). Statistical significance was analyzed by two-sided log-rank (Mantel-Cox) test (B-D, H and K), two-way analysis of variance (ANOVA) (E-G, and I), or two-tailed unpaired Student's *t* test (J). *$P < 0.05$, **$P < 0.01$, ***P < 0.001.

Further, neutrophil extracellular traps digestion in vivo resulted in prolonged survival of mice after the Delta P80 virus infection (Fig 7K). Together, these data suggested that TNF-α-stimulated CXCL1 from macrophages may enhance neutrophil recruitment, lung tissue damage and mortality following the Delta P80 virus infection.

## Inhibition of TNF-α alleviates the Delta P80 virus-associated mortality

We next examined whether inhibition of TNF-α secretion can alleviate the Delta P80 virus-associated mortality. To this end, we first injected mice intravenously or intranasally with TNF protease inhibitor 2 (TAPI-2), a broad-spectrum inhibitor of matrix metalloprotease and TNF converting enzyme, after the Delta P80 virus infection. Intravenous, but not intranasal (S11 Fig), administration of TAPI-2 to mice resulted in improved survival after Delta P80 infection compared to controls (Fig 8A). The levels of pulmonary TNF-α and neutrophil recruitment into the lung tissue were significantly reduced in mice treated with TAPI-2 intravenously, whereas no significant reduction was observed in pulmonary IL-6 and viral load (Fig 8B–8F). Interestingly, the effect of TAPI-2 appears to be more pronounced in aged mice (S12 Fig),

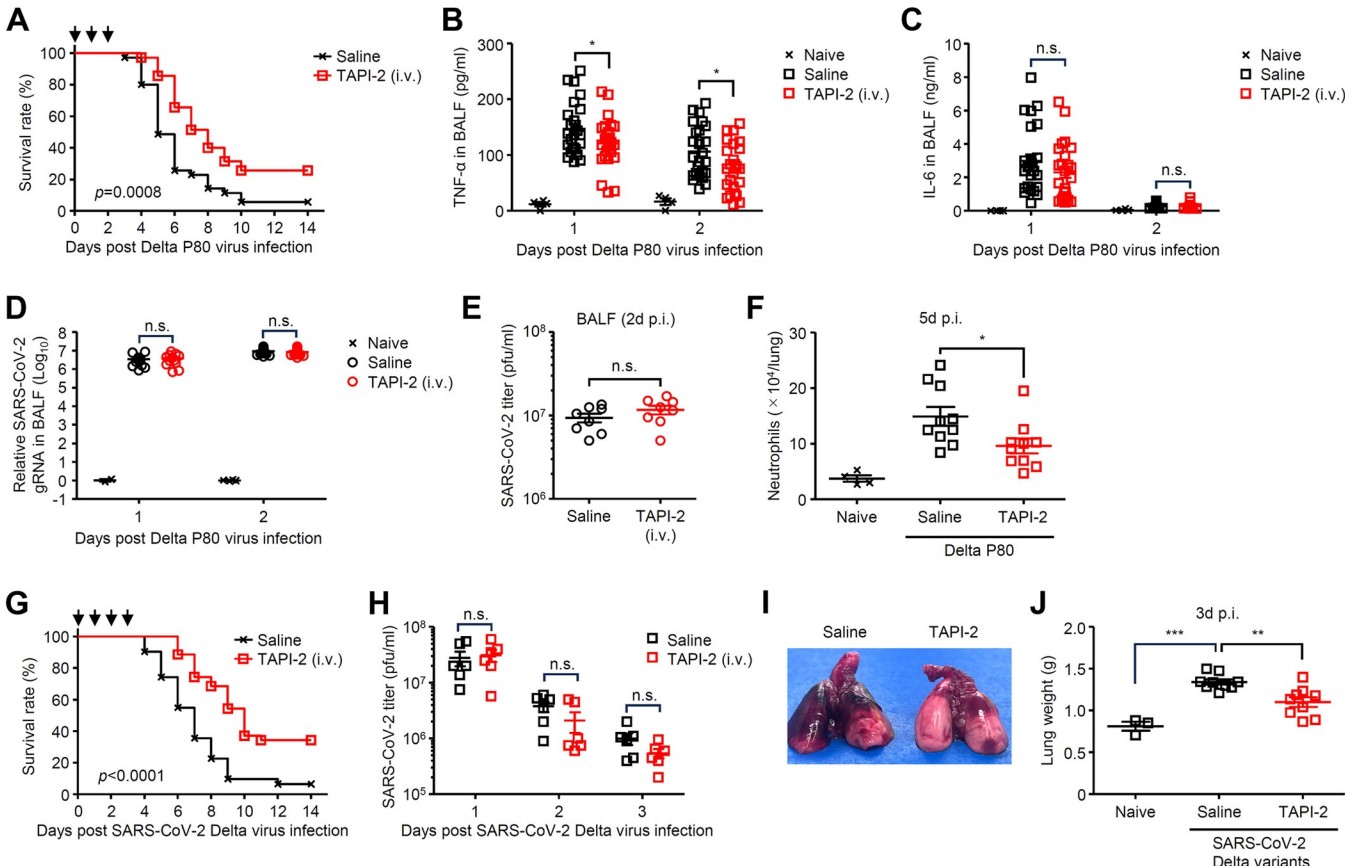

**Fig 8. TAPI-2 alleviates SARS-CoV-2-associated mortality. (A-F)** Six-week-old C57BL/6 mice infected with $1 \times 10^5$ pfu of the Delta P80 virus were administered intravenously with saline or TAPI-2 (2.5 µg) at 0, 1, and 2 days p.i. (allow). Mortality was monitored for 14 days (A). The lung washes were collected at indicated time points and analyzed for TNF-α (B) or IL-6 (C) by ELISA. Total RNAs were extracted from lung washes and SARS-CoV-2 N gRNA levels were assessed by quantitative reverse transcription PCR (D). Viral titers were determined by standard plaque assay (E). Five days later, leukocytes were isolated from the lung. The number of Ly6C$^+$ Ly6G$^+$ neutrophils were analyzed by flow cytometry (F). **(G-J)** Four-week-old Syrian hamsters infected with $8 \times 10^6$ pfu of wild-type SARS-CoV-2 Delta variant were administered intravenously with saline or TAPI-2 (3 µg) at 0, 1, 2, and 3 days p.i. (allow). Mortality was monitored for 14 days (G). The lung washes were collected at indicated time points and viral titers were determined by standard plaque assay (H). Gross lung pathology (I) and total lung weight (J) of hamsters infected with wild-type SARS-CoV-2 Delta variant at 3 days p.i.. Each symbol indicates individual values (B-F, H, and J). Statistical significance was analyzed by two-tailed unpaired Student's *t* test (B, C, D, E, and H), two-sided log-rank (Mantel-Cox) test (A and G), or two-way analysis of variance (ANOVA) (F and J). *$P < 0.05$, **$P < 0.01$, ***$P < 0.001$, n.s., not significant.

probably due to enhanced inflammatory responses in aged mice compared to younger mice (S13 Fig). Further, TAPI-2-treated hamsters had improved survival and pulmonary edema relative to control group following wild-type SARS-CoV-2 Delta variant infection without affecting pulmonary virus titers (Fig 8G–8J). In addition, we examined the effects of TAPI-2 treatment on influenza virus-induced mortality. Following influenza virus infection, the levels of pulmonary TNF-α but not IL-6 were significantly suppressed in TAPI-2-treated group (S14A and S14B Fig). In addition, the TAPI-2-treated mice had improved survival relative to control group following influenza virus infection without affecting pulmonary virus titers (S14C and S14D Fig). Finally, we investigated whether the administration of anti-TNF-α antibodies improves the survival rate of the Delta P80 virus-infected mice. Intravenous, but not intranasal (S15 Fig), administration of anti-TNF-α antibodies to mice significantly reduced pulmonary TNF-α levels and improved survival after the Delta P80 virus infection compared to control group, without affecting pulmonary viral load (S16 Fig). Taken together, our data show that pulmonary TNF-α amplified by type I IFN signals exacerbated the SARS-CoV-2

infection by stimulating CXCL1 production from macrophages. Under such conditions, protease inhibitors that block TNF-α secretion or anti-TNF-α antibodies might be a possible therapeutic drug to reduce the tissue damage and severity of the disease.

## Discussion

The pathogenicity of mouse-adapted SARS-CoV-2 in mice depends on their genetic background, age, and sex [4,17–19]. Thus far, several groups established mouse-adapted SARS-CoV-2 that cause lethal infection in aged Balb/c or C57BL/6 mice but not 6-week-old young C57BL/6 mice [3–6,17–20]. Here we established, for the first time, a mouse-adapted SARS-CoV-2 Delta variant that cause lethal infection in 6-week-old young C57BL/6 mice by serial passaging 80 times in mice. Although the mouse-adapted ancestral SARS-CoV-2 acquired 27 amino acid substitutions throughout the viral genome during 80 serial passages in mice, it did not cause lethal infection in 6-week-old young C57BL/6 mice. Similarly, the mouse-adapted SARS-CoV-2 Delta variants acquired 27 amino acid substitutions throughout the viral genome during 80 serial passages in mice. In contrast to the ancestral P80 virus, the Delta P80 virus caused severe pulmonary edema and lethal infection in 6-week-old young C57BL/6 mice. These data suggest that serial 80 passages do not simply increase the pathogenicity of the mouse adapted SARS-CoV-2 Delta variant, but the Delta P80 virus may maintain the virological characteristics and pathogenicity of the parental SARS-CoV-2 Delta variant during 80 serial passages in mice. It is noteworthy that the nsp9-T35I substitution (Orf1a T4175I), which was identified in the Delta P80 virus, is a characteristic mutation of the currently circulating EG.5 and BA.2.86/JN.1 variants [21,22]. In addition, E-T30I substitution, which was found in the ancestral P80, Delta P18, P29, P41, P61, and P80 viruses, was determined to be the second most frequent recurrently occurring mutation arising in persistent infection [23]. Importantly, a single variant lineage, B.1.616 that is associated with high lethality [24], does contain E-T30I as a lineage-defining mutation [23].

The Delta P80 virus significantly enhanced viral replication in the lung of young C57BL/6 mice compared to the mouse-adapted Delta P18, P29, P41, or P61 variants without affecting viral replication in VeroE6/TMPRSS2 cells. In addition, the Delta P80 virus caused severe pulmonary edema and lethal infection in 6-week-old young C57BL/6 mice. The Delta P80 virus acquired 7 new amino acid substitutions during an additional 19 passages from the P61 virus. These 7 amino acid substitutions include nsp2, nsp3, S, and M genes. Since multiple SARS-CoV-2 proteins including nsp2, nsp3, and M inhibit *Ifnb1* transcription [25,26], these amino acid substitutions in nsp2, nsp3, and M proteins of the Delta P80 virus could contribute to inhibit host interferon responses and efficient viral replication *in vivo*. In this study, we showed that treatment with recombinant IFN-α/β at the time of infection completely protected mice from lethal Delta P80 virus infection. These results suggest that host interferon responses at the time of infection are important for suppressing SARS-CoV-2 replication *in vivo*. It has been reported that among patients with severe COVID-19, some have deficiencies in toll-like receptor 3 (TLR3), TLR7, or interferon (IFN) signaling, or possess autoantibodies against type I IFNs [27–29]. However, our data also showed that intranasal administration to mice with recombinant IFN-α/β at 2 days p.i. exacerbated the ancestral P80 virus infection. In addition, the pulmonary virus titers were comparable between WT and IFNAR1 KO mice. These results indicate that host type I IFN signals do not control a lethal dose ($1 \times 10^5$ pfu) of the Delta P80 virus infection and exacerbate the disease severity. The discrepancies in these results may be explained by the timing of type I IFN responses or the differences in infectious doses across the experimental condition. Indeed, we observed no significant differences in body weight change or survival rate between MyD88-deficient and WT mice following a sublethal dose

($1\times10^4$ pfu) of the Delta P80 virus infection. In contrast, MyD88 and IFNAR1 KO mice exhibited resistance to a lethal dose ($1\times10^5$ pfu) of the Delta P80 virus infection. In a lethal dose ($3\times10^4$ pfu) of SARS-CoV infection, Channappanavar and colleagues have demonstrated that the disease severity is ameliorated in *Ifnar*$^{-/-}$ BALB/c mice [30]. These observations suggest that host IFNs can block SARS-CoV-2 infections when viral burdens are low, such as during a physiological dose infection in humans or when mice are infected with the ancestral P80 virus, as demonstrated in this study. However, if innate antiviral immune system deficiencies allow low viral burdens to overcome host antiviral defenses—for example, in individuals with deficiencies in innate immune signals or those with autoantibodies against type I IFNs—or if infections involve high viral loads, as seen in this study where a lethal dose ($1\times10^5$ pfu) was administered intranasally to mice, the IFN response becomes insufficient to control initial viral replication. This insufficiency may lead to excessive inflammation and lung injury. [31]. Recent studies have demonstrated that SARS-CoV-2 induces delayed type I IFN responses [26,32,33]. In addition, the induction of type I IFN in response to viral infection may be impaired in older hosts [34]. Together, the virological characteristics, host interferon responses, and infectious doses may have important implications for SARS-CoV-2 pathogenesis [31,35,36]. Future studies using physiological doses of mouse-adapted SARS-CoV-2 variants in aged, immunocompromised, or obese mice will facilitate the development of efficacious interventions and treatments for severe COVID-19.

Our data have demonstrated that MyD88 or IFNAR1 KO mice are highly resistant to lethal the Delta P80 virus infection compared with WT mice. Pulmonary virus titers were comparable between WT and MyD88 or IFNAR1 KO mice. Instead, the levels of IFN-α, IFN-β, IFN-λ, and TNF-α were significantly reduced in the lungs of MyD88 and IFNAR1 KO mice compared with those of WT mice. Consistent with a previous report [37], we found that treatment of bone marrow-derived macrophages with TNF-α significantly stimulated CXCL1 production. In addition, we showed that intranasal administration of CXCL1 or TNF-α at 2 days p.i. exacerbated the ancestral P80 virus infection. CXCL1 is a well-known major chemoattractant of neutrophils. It has increasingly become evident that excess neutrophil recruitment into the lung exacerbates SARS-CoV-2 infection [10–14]. In fact, we demonstrated that nasal administration of CXCL1 enhanced neutrophil recruitment into the lung. In addition, previous studies have demonstrated that combination of TNF-α and IFN-γ synergistically induce inflammatory cell death and exacerbates SARS-CoV-2-induced mortality [5,38]. In addition to inflammatory role of TNF-α in severity of SARS-CoV-2 infection, TNF-α is known to cause bronchial hyperreactivity, narrowing of the airways, damage to the respiratory epithelium, stimulation of collagen synthesis and fibrosis in the respiratory system [39]. Recently, it has demonstrated that aged TNF KO mice exhibit resistance to lethal infection with SARS-CoV-2 N501Y P21 virus infection without affecting virus titers in the lung compared with aged WT mice [3]. Similarly, we found that intravenous treatment of the Delta P80 virus-infected mice with an anti-TNF-α antibody significantly improved survival relative to control group without affecting pulmonary virus titers. Together, these data suggest that type I IFNs signals may exacerbate SARS-CoV-2 infection by stimulating TNF-α-induced neutrophils recruitment and/or TNF-α/IFN-γ-induced inflammatory cell death in the lung tissue. Thus, TNF-α could be considered a potential therapeutic target for severe COVID-19 [40].

It is difficult to directly compare the relative virulence of the ancestral and delta variants in humans, given the potential influence of vaccination on the disease severity. A previous study demonstrated that the Delta variant is more virulent than the ancestral strain in dwarf hamsters [41]. Consistent with this observation, we showed that the Delta P80 virus exhibited a higher mortality rate in young C57BL/6 mice compared to the ancestral P80 virus. There are several possible explanations for how specific mutations observed in the Delta P80 virus

contribute to MyD88/IFNAR1-mediated severe disease outcomes. First, the Delta P80 virus retained mutations characteristic of the Delta variant spike protein, specifically L452R and P681R, which are important for increasing the fusogenicity of the spike protein and the formation of syncytia in infected cells [42,43]. Syncytial death via apoptosis, pyroptosis, or TNF-mediated necroptosis can release the pathogen-associated molecular patterns (PAMPs) and enhance excessive inflammatory responses at the site of infection [44]. Second, interferon induced transmembrane protein 1 (IFITM1), induced in a MyD88 and IFNAR1 signaling dependent manner [45], is involved in the inhibition of syncytium formation in SARS-CoV-2 infected cells [46]. It is possible that specific mutations observed in the Delta P80 virus may contribute to inhibiting MyD88/IFNAR1-mediated IFITM1 expression [31], which could potentially lead to an increase in syncytium formation in SARS-CoV-2-infected cells and an exacerbation of the syncytial death-mediated inflammatory response. Third, the ancestral and Delta P80 viruses primarily target CD45.2-negative epithelial cells in the lung, but the specific mutations observed in the Delta P80 virus spike protein may have resulted in an increased infection rate of alveolar macrophages and dendritic cells in the lung tissue. This may potentially result in a MyD88/IFNAR1-dependent excessive inflammatory responses and the development of severe pulmonary edema in young C57BL/6 mice. However, treatment of mice with TAPI-2, anti-TNF-α antibody, or DNase I had a limited, but significant, effect on increasing survival after the Delta P80 virus infection. Therefore, the identification of more efficacious methods for the suppression of TNF-α production or neutrophil recruitment may facilitate the development of superior therapeutic interventions for the treatment of severe cases of COVID-19.

In summary, our findings substantially expand our understanding of how innate antiviral immune signals exacerbate SARS-CoV-2 infection. In addition, we established, for the first time, a mouse-adapted SARS-CoV-2 Delta variant that cause lethal infection in 6-week-old young C57BL/6 mice by serial passaging 80 times in mice. The Delta P80 virus infection enhanced pulmonary proinflammatory cytokines production including TNF-α in a MyD88- and IFNAR1-dependent manner. The TNF-α stimulated CXCL1 production from bone marrow-derived macrophages, which may enhance lung tissue damage and the disease severity following the Delta P80 virus infection (S17 Fig). Because decreased IFN and elevated proinflammatory cytokines are a common characteristic of the innate immune system in older humans [34], our results imply a possible effect of SARS-CoV-2-induced proinflammatory cytokines in severity of the diseases in elderly people. In addition, it will be important to determine whether type I IFNs therapy can be used for severe COVID-19.

## Materials and methods

### Ethics statement

All experiments with SARS-CoV-2 were performed in enhanced biosafety level 3 (BSL3) containment laboratories at the University of Tokyo, in accordance with the institutional biosafety operating procedures. All animal experiments including generation of mouse-adapted SARS-CoV-2 variants were performed in accordance with University of Tokyo's Regulations for Animal Care and Use, which were approved by the Animal Experiment Committee of the Institute of Medical Science, the University of Tokyo (PA22-33).

### Animals

Six-week-old female C57BL/6JJmsSlc, BALB/cCrSlc, C3H/HeYokSlc mice, and 4-week-old female Syrian hamsters obtained from Japan SLC, Inc. were used as WT controls. For some experiments we used aged (64- to 83-week-old) female C57BL/6JJcl mice obtained from CLEA

Japan, Inc. MyD88-deficient C57BL/6 mice were purchased from Oriental Bioservice (Kyoto, Japan). IFNAR1-deficient C57BL/6 mice were described previously [47].

## Cell culture

VeroE6 cells stably expressing transmembrane protease serine 2 (VeroE6/TMPRSS2; JCRB Cell Bank 1819) were maintained in Dulbecco's modified Eagle's medium (DMEM) (low-glucose) (Nacalai Tesque, 08456–65) supplemented with 10% v/v fetal bovine serum (FBS), 1% v/v penicillin (100 units/ml)/streptomycin (100 μg/ml), and G418 (1 mg ml$^{-1}$; Nacalai Tesque, 16512–94). L929 cells were maintained in DMEM (high-glucose) (Nacalai Tesque, 08458–45) supplemented with 10% v/v FBS and 1% v/v penicillin/streptomycin (P/S). To prepare bone marrow-derived macrophages, bone marrows from the tibia and femur were obtained by flushing with DMEM. Bone marrow cells were cultured with DMEM supplemented with 10% FBS, L-glutamine, 1% P/S, and 30% L929 supernatant containing the macrophage colony-stimulating factor at 37°C for 5 days [48].

## Viruses

An ancestral SARS-CoV-2 strain bearing aspartic acid at position 614 of spike (S) protein (S-614D) [8] and the Delta variant hCoV-19/Japan/TY11-927-P1/2021 (lineage B.1.617.2, GISAID ID: EPI_ISL_2158617) [9] variant were grown in VeroE6/TMPRSS2 cells for 2 days at 37°C. Viral titers were quantified by a standard plaque assay using VeroE6/TMPRSS2 cells and viral stock was stored at -80°C.

For generation of mouse-adapted SARS-CoV-2 variants, 64- to 83-week-old C57BL/6 mice were intranasally infected with an ancestral SARS-CoV-2 (SARS-CoV-2/ UT-NCGM02/human/2020) [8] or the Delta variant hCoV-19/Japan/TY11-927-P1/2021 (lineage B.1.617.2, GISAID ID: EPI_ISL_2158617) [9]. The lung washes were collected at 3 days p. i.. Then VeroE6/TMPRSS2 cells were inoculated with the lung washes to propagate the mouse-adapted SARS-CoV-2 (refer to as P1) (Fig 1A). After 5 serial passages in aged mice, 6-weeks-old young C57BL/6 mice were intranasally infected with the P5 virus (Fig 1A). The lung washes were collected at 3 days p.i. and VeroE6/TMPRSS2 cells were inoculated with the lung washes to propagate the P6 virus (Fig 1A). After 10 serial passages in young mice, we serially passaged the lung washes of the virus-infected young mice every 3 days without virus propagation in VeroE6/TMPRSS2 cells (Fig 1A). The ancestral and the Delta P80 viral stocks used in the experiments were grown in VeroE6/TMPRSS2 cells for 2 days at 37°C. Viral titers were quantified by a standard plaque assay using VeroE6/TMPRSS2 cells and viral stock was stored at -80°C.

A mouse-adapted influenza A virus strain A/Puerto Rico/8/1934 (PR8) was grown in allantoic cavities of 10-d-old fertile chicken egg for 2 days at 35°C. Viral titers were quantified by a standard plaque assay using MDCK cells and viral stock was stored at –80°C.

For intranasal infection, mice were infected by intranasal application of 50 μL of virus suspension ($1\times10^5$ pfu of mouse-adapted SARS-CoV-2 variants or $1\times10^3$ pfu of PR8 in PBS) under isoflurane anaesthesia. Syrian hamsters were infected by intranasal application of 400 μL of virus suspension ($8\times10^6$ pfu of wild-type SARS-CoV-2 Delta variant in PBS).

## Reagents

A recombinant mouse IFN-α (Cat#HC1040-10) was purchased from Hycult Biotech. A recombinant mouse IFN-β (Cat#8234-MB-010) was obtained from R&D Systems. Recombinant mouse CXCL1 (Cat#250–11) and TNF-α (Cat#315-01A) were from PeproTech. TAPI-2 (Cat#INH-3852-PI) was purchased from Biosynth. Monoclonal antibody against mouse TNF-

α (XT3.11, Cat#BE0058) and rat IgG1 isotype control (HRPN, Cat#BE0088) were purchased from Bio X Cell.

## Quantitative PCR

Total RNA was extracted from lung washes using TRIzol reagent (Invitrogen, 15596018) and reverse transcribed into cDNA using SuperScript III reverse transcriptase (Invitrogen, 18080085) with a SARS-CoV-2 N reverse primer (5'- tctggttactgccagttgaatctg-3'). TB Green Premix Ex Taq II (TaKaRa, RR820A) and a LightCycler 1.5 instrument (Roche Diagnostics) were used for quantitative PCR with the following primers: SARS-CoV-2 N forward, 5'-gaccccaaaatcagcgaaat-3', and reverse, 5'-tctggttactgccagttgaatctg-3'; influenza virus NP forward, 5'-agaacatctgacatgaggac-3', and reverse, 5'-gtcaaaggaaggcacgatc-3' [16,49].

## ELISA

Cell-free supernatants or lung washes were analyzed for the presence of IFN-α (Hycult Biotech, HM1001; PBL Assay Science, 32100–1), IL-1β (eBioscience, 14-7012-85 and 13-7112-85), IL-6 (eBioscience, 14-7061-85 and 13-7062-85), and TNF-α (eBioscience, 14-7423-85 and 13-7341-85) using an enzyme-linked immunosorbent assay (ELISA) utilizing paired antibodies [50]. IFN-β (PBL Assay Science, 42400–1), IFN-λ (PBL Assay Science, 62830–1), or CXCL1 (Proteintech, KE10019) ELISA was performed according to the manufacturer's instructions. Absorbance at 450 nm was measured by using Microplate Manager version 6 (Bio-Rad).

## Flow cytometry

The single-cell suspensions of lung samples were prepared as previously described [51]. Briefly, lungs were perfused with 10 ml PBS through the right ventricle, minced using razor blades, and incubated in HBSS containing 2.5 mM Hepes and 1.3 mM EDTA at 37°C for 30 min. The cells were resuspended in RPMI containing 5% FBS, 1 mM $CaCl_2$, 1 mM $MgCl_2$, 2.5 mM Hepes, and 0.5 mg/ml collagenase D (Roche) and incubated at 37°C for 60 min. A single-cell suspension was prepared after red blood cell lysis. The resulting cells were filtered through a 70-μm cell strainer (BD). For neutrophil staining, cells were incubated with APC-labeled anti-Ly6G (Invitrogen, 17-9668-82; 1:200) and eFluor 450-labeled anti-Ly6C (Invitrogen, 48-5932-82; 1:200) (S18 Fig). For the detection of SARS-CoV-2-infected cells, cells were fixed and permeabilized using a Cytofix/Cytoperm kit (BD Biosciences, 554714), and intracellulary stained with PE-labeled rabbit anti-SARS-CoV-2 nucleocapsid (abcam, ab283244; 1:200) antibody. Flow cytometric analysis was performed with a FACSVerse flow cytometer (BD Biosciences). The final analysis and graphical output were performed using FlowJo software (Tree Star, Inc.).

## Statistical analysis

Statistical significance was tested using nonparametric one-way analysis of variance (ANOVA) with Tukey's multiple comparison test, non-parametric Mann-Whitney t test, or Student's two-tailed, unpaired t test where indicated in the figure legend, using PRISM software (version 5; GraphPad software). $P < 0.05$ was considered statistically significant.

## Supporting information

**S1 Fig. Neutrophil recruitment into the lung after the ancestral or Delta P80 virus infection.** Six-week-old C57BL/6 mice were infected intranasally with $1×10^5$ pfu of the ancestral or Delta P80 virus. Five days later, leukocytes were isolated from the lung. The number of $Ly6C^+$

Ly6G$^+$ neutrophils were analyzed by flow cytometry. Each symbol indicates individual values. Statistical significance was analyzed by two-way analysis of variance (ANOVA). **P $<$ 0.01. (TIF)

**S2 Fig. Gating strategy to assess the frequency of the SARS-CoV-2-infected CD45.2$^+$ or CD45.2$^−$ cells.** Six-week-old C57BL/6 mice were infected intranasally with 1×10$^5$ pfu of the ancestral or Delta P80 virus. Leukocytes were isolated from the lung at 3 days post infection, and intracellularly stained with the nucleoprotein-specific antibody. The frequency of the SARS-CoV-2-infected CD45.2$^+$ or CD45.2$^−$ cells were analyzed by flow cytometry. (TIF)

**S3 Fig. Gating strategy to assess the frequency of the SARS-CoV-2-infected alveolar macrophages. (A and B)** Six-week-old C57BL/6 mice were infected intranasally with 1×10$^5$ pfu of the ancestral or Delta P80 virus. Leukocytes were isolated from the lung washes at 3 days post infection, and intracellularly stained with the nucleoprotein-specific antibody. To identify alveolar macrophage population, leucocytes were gated by forward and side scatter. Doublet signals are excluded by plotting forward scatter area versus forward scatter height. CD11b and CD11c double-positive cells were identified as alveolar macrophages (A). Frequency of nucleoprotein$^+$ cells in alveolar macrophages are shown (B). **(C)** The population was confirmed to be macrophages by depletion with clodronate liposomes. Each symbol indicates individual values. Statistical significance was analyzed by two-way analysis of variance (ANOVA) (B). ***P $<$ 0.001. (TIF)

**S4 Fig. Gating strategy to assess the frequency of the SARS-CoV-2-infected dendritic cells in the lung. (A and B)** Six-week-old C57BL/6 mice were infected intranasally with 1×10$^5$ pfu of the ancestral or Delta P80 virus. Leukocytes were isolated from the lung at 3 days post infection, and intracellularly stained with the nucleoprotein-specific antibody. To identify dendritic cell population, leucocytes were gated by forward and side scatter. Doublet signals are excluded by plotting forward scatter area versus forward scatter height. CD11c-positive cells were identified as dendritic cells (A). Frequency of nucleoprotein$^+$ cells in CD11c$^+$ dendritic cells are shown (B). Each symbol indicates individual values. Statistical significance was analyzed by two-way analysis of variance (ANOVA) (B). ***P $<$ 0.001, n.s., not significant. (TIF)

**S5 Fig. Gating strategy to assess the frequency of the SARS-CoV-2-infected B cells in the lung. (A and B)** Six-week-old C57BL/6 mice were infected intranasally with 1×10$^5$ pfu of the ancestral or Delta P80 virus. Leukocytes were isolated from the lung at 3 days post infection, and intracellularly stained with the nucleoprotein-specific antibody. To identify B cell population, leucocytes were gated by forward and side scatter. Doublet signals are excluded by plotting forward scatter area versus forward scatter height. CD19 and B220 double-positive cells were identified as B cells (A). Frequency of nucleoprotein$^+$ cells in B cells are shown (B). Each symbol indicates individual values. Statistical significance was analyzed by two-way analysis of variance (ANOVA) (B). ***P $<$ 0.001, n.s., not significant. (TIF)

**S6 Fig. Gating strategy to assess the frequency of the SARS-CoV-2-infected CD4$^+$ or CD8$^+$ T cells in the lung. (A-C)** Six-week-old C57BL/6 mice were infected intranasally with 1×10$^5$ pfu of the ancestral or Delta P80 virus. Leukocytes were isolated from the lung at 3 days post infection, and intracellularly stained with the nucleoprotein-specific antibody. To identify CD4$^+$ and CD8$^+$ T cell population, leucocytes were gated by forward and side scatter. Doublet

signals are excluded by plotting forward scatter area versus forward scatter height. CD3 and CD4 or CD3 and CD8 double-positive cells were identified as CD4$^+$ and CD8$^+$ T cells, respectively (A). Frequency of nucleoprotein$^+$ cells in CD4$^+$ (B) and CD8$^+$ T cells (C) are shown. Each symbol indicates individual values. Statistical significance was analyzed by two-way analysis of variance (ANOVA) (B and C). **$P < 0.01$, ***$P < 0.001$, n.s., not significant.
(TIF)

**S7 Fig. Gating strategy to assess the frequency of the SARS-CoV-2-infected neutrophils in the lung. (A and B)** Six-week-old C57BL/6 mice were infected intranasally with $1 \times 10^5$ pfu of the ancestral or Delta P80 virus. Leukocytes were isolated from the lung at 3 days post infection, and intracellularly stained with the nucleoprotein-specific antibody. To identify neutrophil population, leucocytes were gated by forward and side scatter. Doublet signals are excluded by plotting forward scatter area versus forward scatter height. Then, B cells and T cells were excluded based on B220 and CD3 expression, respectively. Ly6C and Ly6G double-positive cells were identified as neutrophils (A). Frequency of nucleoprotein$^+$ cells in neutrophils are shown (B). Each symbol indicates individual values. Statistical significance was analyzed by two-way analysis of variance (ANOVA) (B). *$P < 0.05$, ***$P < 0.001$.
(TIF)

**S8 Fig. Effects of MyD88 deficiency on body weight changes and survival after sublethal dose of Delta P80 virus infection. (A and B)** Six-week-old C57BL/6 WT or MyD88 mice were infected intranasally with $1 \times 10^4$ pfu of the Delta P80 virus. Weight loss (A) and mortality (B) were monitored for 14 days. Statistical significance was analyzed by two-tailed unpaired Student's *t* test (A) or two-sided log-rank (Mantel-Cox) test (B).
(TIF)

**S9 Fig. Infection of mice with Delta P80 virus does not stimulate detectable levels of IL-1β in BALF.** C57BL/6 WT, MyD88, or IFNAR1 KO mice were infected intranasally with $1 \times 10^5$ pfu of the Delta P80 virus. The lung washes were collected at 2 days p.i. and analyzed for IL-1β by ELISA. Each symbol indicates individual values.
(TIF)

**S10 Fig. TNF-α exacerbates Delta P80 virus infection in MyD88 mice.** Six-week-old MyD88 KO mice infected with the Delta P80 virus were administered intranasally with PBS or recombinant mouse TNF-α (2.5 μg) at 2 days p.i. (arrow). Mortality was monitored for 14 days. Statistical significance was analyzed by two-sided log-rank (Mantel-Cox) test.
(TIF)

**S11 Fig. Intranasal administration of TAPI-2 has no effect on the survival rate of the Delta P80 virus-infected mice.** Six-week-old C57BL/6 mice infected with $1 \times 10^5$ pfu of the Delta P80 virus were administered intranasally with saline or TAPI-2 (0.5 μg) at 1 and 2 days p.i. (allow). Mortality was monitored for 14 days. Statistical significance was analyzed by two-sided log-rank (Mantel-Cox) test.
(TIF)

**S12 Fig. TNF protease inhibitor 2 alleviates SARS-CoV-2-accosiated mortality in aged mice. (A and B)** Aged (21-week-old) C57BL/6 mice were infected intranasally with 100 pfu of the ancestral P80 virus. Then, infected mice were administered intravenously with saline or TAPI-2 (2.5 μg) at indicated time points (arrows). Mortality was monitored for 14 days (A). The lung washes were collected at 3 days p.i. and viral titers were determined by standard plaque assay (B). Each symbol indicates individual values (B). Statistical significance was analyzed by two-sided log-rank (Mantel-Cox) test (A), or two-tailed unpaired Student's t test (B). n.s.,

not significant.
(TIF)

**S13 Fig. TNF protease inhibitor 2 suppresses SARS-CoV-2-induced inflammatory responses in aged mice. (A-C)** Aged (55-week-old) C57BL/6 mice were infected intranasally with $1\times10^5$ pfu of the Delta P80 virus. Then, infected mice were administered intravenously with saline or TAPI-2 (2.5 μg) at 0, 1, and 2 days p.i.. The lung washes were collected at 2 days p.i. and analyzed for TNF-α (A), IL-6 (B), or CXCL1 (C) by ELISA. Each symbol indicates individual values. Statistical significance was analyzed by two-way analysis of variance (ANOVA). ***P < 0.001, n.s., not significant.
(TIF)

**S14 Fig. TAPI-2 alleviates influenza virus-associated mortality in mice.** Six-week-old C57BL/6 mice infected with $1\times10^3$ pfu of the PR8 virus were administered intravenously with saline or TAPI-2 (2.5 μg) at 0, 1, and 2 days p.i. (allow). **(A-C)** The lung washes were collected at 2 days p.i. and analyzed for TNF-α (A) or IL-6 (B) by ELISA. Total RNAs were extracted from lung washes and influenza virus NP RNA levels were assessed by quantitative reverse transcription PCR (C). **(D)** Mortality was monitored for 14 days. Each symbol indicates individual values (A-C). Statistical significance was analyzed by two-tailed unpaired Student's t test (A-C), or two-sided log-rank (Mantel-Cox) test (D). ***P < 0.001, n.s., not significant.
(TIF)

**S15 Fig. Intranasal administration of anti-TNF-α antibodies has no effect on the survival rate of the Delta P80 virus-infected mice.** Six-week-old C57BL/6 mice infected with $1\times10^5$ pfu of the Delta P80 virus were administered intranasally with isotype rat IgG (2.5 μg) or anti-TNF-α antibodies (2.5 μg) at 1 day p.i. (allow). Mortality was monitored for 14 days. Statistical significance was analyzed by two-sided log-rank (Mantel-Cox) test.
(TIF)

**S16 Fig. Intravenous administration of anti-TNF-α antibodies alleviates Delta P80 virus-associated mortality. (A-C)** Six-week-old C57BL/6 mice infected with $1\times10^5$ pfu of the Delta P80 virus were administered intravenously with isotype rat IgG (10 μg) or anti-TNF-α antibodies (10 μg) at 1-day p.i. (allow). Mortality was monitored for 14 days (A). The lung washes were collected at 2 days p.i. and analyzed for TNF-α by ELISA (B). Total RNAs were extracted from lung washes and SARS-CoV-2 N gRNA levels were assessed by quantitative reverse transcription PCR (C). Each symbol indicates individual values (B and C). Statistical significance was analyzed by two-sided log-rank (Mantel-Cox) test (A), two-way analysis of variance (ANOVA) (B), or two-tailed unpaired Student's t test (C). ***P < 0.001, n.s., not significant.
(TIF)

**S17 Fig. Proposed mechanism by which type I IFN signals exacerbate SARS-CoV-2 infection.** Infection with a lethal dose ($1\times10^5$ pfu) of the Delta P80 virus enhances type I IFNs and proinflammatory cytokines production in a MyD88- and IFNAR1-dependent manner. TNF-α stimulates CXCL1 production from macrophages, which may enhance lung tissue damage by neutrophils and the disease severity following the Delta P80 virus infection.
(TIF)

**S18 Fig. Gating strategy for identifying neutrophils in the lung.** To identify neutrophil population, leucocytes were gated by forward and side scatter. Doublet signals are excluded by plotting forward scatter area versus forward scatter height. Then, B cells and T cells were excluded based on B220 and CD3 expression, respectively. Ly6C and Ly6G double-positive

cells were identified as neutrophils.
(TIF)

**S1 Data. Raw data of main figures.**
(XLSX)

**S2 Data. Raw data of supporting information figures.**
(XLSX)

## Acknowledgments

We thank Yoshihiro Kawaoka (University of Wisconsin and University of Tokyo) for providing SARS-CoV-2/UT-NCGM02/Human/2020/Tokyo, Ken Maeda (National Institute of Infectious Diseases) for providing SARS-CoV-2 Delta variant, Takara Bio Inc. and Rhelixa Inc. for next-generation sequencing analysis. Flow cytometric analysis was performed in the IMSUT FACS Core laboratory.

## Author Contributions

**Conceptualization:** Takeshi Ichinohe.

**Funding acquisition:** Shinji Fukuda, Kensuke Miyake, Takeshi Ichinohe.

**Investigation:** Moe Kobayashi, Nene Kobayashi, Kyoka Deguchi, Seira Omori, Minami Nagai, Ryutaro Fukui, Isaiah Song, Takeshi Ichinohe.

**Methodology:** Shinji Fukuda, Kensuke Miyake, Takeshi Ichinohe.

**Project administration:** Takeshi Ichinohe.

**Resources:** Ryutaro Fukui, Shinji Fukuda, Kensuke Miyake, Takeshi Ichinohe.

**Supervision:** Shinji Fukuda, Kensuke Miyake, Takeshi Ichinohe.

**Validation:** Moe Kobayashi, Nene Kobayashi, Shinji Fukuda, Kensuke Miyake, Takeshi Ichinohe.

**Writing – original draft:** Takeshi Ichinohe.

**Writing – review & editing:** Moe Kobayashi, Ryutaro Fukui, Shinji Fukuda, Kensuke Miyake, Takeshi Ichinohe.

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
