## [Decision Letter · Decision Letter 0]

26 Jul 2024

Dear Dr. Ichinohe,

Thank you very much for submitting your manuscript "TNF-α exacerbates SARS-CoV-2-infection by stimulating CXCL-1 production from macrophages" for consideration at PLOS Pathogens. As with all papers reviewed by the journal, your manuscript was reviewed by members of the editorial board and by several independent reviewers. In light of the reviews (below this email), we would like to invite the resubmission of a significantly-revised version that takes into account the reviewers' comments.

We cannot make any decision about publication until we have seen the revised manuscript and your response to the reviewers' comments. Your revised manuscript is also likely to be sent to reviewers for further evaluation.

Sincerely,

Jie Sun, Ph.D.

Academic Editor

PLOS Pathogens

Sonja Best

Section Editor

PLOS Pathogens

Michael Malim

Editor-in-Chief

PLOS Pathogens

orcid.org/0000-0002-7699-2064

Reviewer's Responses to Questions

**Part I - Summary**

Reviewer #1: Deguchi et al's manuscript "TNF-α exacerbates SARS-CoV-2-infection by stimulating CXCL-1 production from macrophages" describes 2 mouse adapted SARS-CoV-2 infection models and investigates the pathogenesis of SARS-CoV-2 infection based on these models. Their work provides potential tools for further identify the pathology of COVID-19. However, some substantial concerns need to be addressed.

Reviewer #2: This manuscript titled “TNF exacerbates SARS-CoV-2 infection by stimulating CXCL-1 production from macrophages” by Deguchi K et al. has two primary objectives: a) to develop a reliable mouse-adapted (MA)-virus that causes severe disease in commonly used young C57BL/6 mice, and b) to investigate the basis for SARS-CoV-2-induced severe disease using the novel MA virus. Here, the authors examine underlying basis for SARS-CoV-2-induced cytokine storm and severe disease using MA SARS-CoV-2 viruses derived from different human variants. The authors show that high-passaged (p80) MA SARS-CoV-2 derived from delta variant, but not the ancestral variant, caused lethal disease in young B6 mice, while P80 virus from both backgrounds caused severe disease in BALB/c and C3H mice. In young B6 mice, the P80 MA-delta virus replicated to high titers, caused lung pathology, and triggered a robust cytokine response. The authors also showed that Myd88 and IFN-I signaling were pathogenic, early IFN-β treatment protected, and the delayed IFN-β administration caused pathology in P80 MA-delta virus-infected B6 mice. Mechanistically, the IFN-I-induced TNF-mediated CXCL-1 response was associated with severe disease in P80 MA-delta virus-infected young B6 mice.

Strengths:

This is an interesting study wherein the authors have developed a novel MA virus on the Delta variant background for studying SARS-CoV-2 pathogenesis in B6 mice. The study also highlights the SARS-CoV-2 variant-specific role of IFN-I and Myd88 in cytokine storm and severe disease. Additionally, the work identifies the role of the TNF-CXCL-1 axis in causing severe lung inflammation and lethal pneumonia. However, there are several major concerns that diminish the enthusiasm for this work in its current form. Comments are listed below.

Weaknesses:

1. Study implications: It is now well-established that the lack of IFN-I and TLR3/7 signaling is associated with severe COVID19. As a result, significance and implications of the differential role of IFN-I and Myd88 in MA-ancestral and MA-delta virus-infected mice, and how these results correlate with COVID-19 outcomes in humans, are not clear, nor are well articulated. Specifically, it is unclear whether the loss of IFN and Myd88 signaling is associated with less severe disease in humans infected with the Delta variant.

2. MA viruses developed by different laboratories (including MA-10 by Baric lab and MA-30 by Perlman lab) are extensively used to study SARS-CoV-2 pathogenesis. Although these viruses cause mild disease in young 6-week B6 mice, these viruses (specifically MA30) cause lethal disease in adult (16-20 week or older) B6 mice. Therefore, the premise of having to develop a MA virus specifically to induce severe disease in 6-week mice is not justified. Additionally, 6-week mice, although extensively used, are too young for SARS-CoV-2 studies, a virus that causes mild disease in young individuals.

3. The differential disease outcomes following MA-ancestral and MA-delta virus infections are novel. However, it would be interesting to know if specific mutations observed in MA-delta contribute to IFN-I/Myd88 mediated severe disease outcomes.

4. The protective and detrimental roles of early and delayed IFN-β treatment, respectively, are well described for SARS-CoV, MERS-CoV, and SARS-CoV-2 infection by several investigators, making these observations less novel. However, the role of IFN-I mediated TNF signaling in CXCL-1-induced lung pathology is novel and significant.

5. Figure 7: CXCL-1 and TNF treatment enhanced disease severity in ancestral and delta P80 MA virus-infected mice. However, TAPI treatment marginally, albeit significantly, enhanced survival. These results show that perhaps using knockout mice or blocking TNF and CXCL-1 using specific monoclonal antibodies is a better approach compared to using an inhibitor.

6. The authors postulate that TNF-mediated CXCL-1-induced neutrophils cause inflammation and pathology in delta-MA infected mice. However, no neutrophil data (FACS or histology) is available to support these conclusions.

**Part II – Major Issues: Key Experiments Required for Acceptance**

Reviewer #1: Major concerns

1 In this work, SARS-CoV-2 viruses were adapted in murine system by serial passages, which is expected to lead to new mutations as shown in Fig. 1 and 3. Authors need to provide further interpretations on these mutations, especially their clinical relevance. If these mutations are identified in SARS-CoV-2 variants circulating in human population as well, the value of these models will be significantly improved.

2 In the pathogenesis investigation, ancestral P80 and delta P80 were used as moderate and lethal models, respectively, which need to be clarified to avoid confusions. Based on the severity caused by different viruses, the role of MyD88 and IFNAR1, as well as different treatment of type I IFN were used correspondingly. Again, these designs are understandable, but the generated data need to be interpreted separately because the pathology of COVID-19 caused by different variants, especially those carrying multiple mutations, could be distinct. In addition, the authors are encouraged to establish different severity model using the same virus by optimizing the infectious doses.

3 The author claim the critical role of CXCL-1 in TNF-induced inflammation but failed to provide direct evidence. In vivo TNF-dependent CXCL-1 expression by macrophage is warranted. Meanwhile, the underlying mechanisms contributing to CXCL-1-mediated inflammation (by recruiting neutrophil as shown in Fig. s4? If so, the neutrophil accumulation in affected lung need to be determined) need to be further identified.

Reviewer #2: 1. The differential disease outcomes following MA-ancestral and MA-delta virus infections are novel. However, it would be interesting to know if specific mutations observed in MA-delta contribute to IFN-I/Myd88 mediated severe disease outcomes.

2. The protective and detrimental roles of early and delayed IFN-β treatment, respectively, are well described for SARS-CoV, MERS-CoV, and SARS-CoV-2 infection by several investigators, making these observations less novel. However, the role of IFN-I mediated TNF signaling in CXCL-1-induced lung pathology is novel and significant.

3. Figure 7: CXCL-1 and TNF treatment enhanced disease severity in ancestral and delta P80 MA virus-infected mice. However, TAPI treatment marginally, albeit significantly, enhanced survival. These results show that perhaps using knockout mice or blocking TNF and CXCL-1 using specific monoclonal antibodies is a better approach compared to using an inhibitor.

4. The authors postulate that TNF-mediated CXCL-1-induced neutrophils cause inflammation and pathology in delta-MA infected mice. However, no neutrophil data (FACS or histology) is available to support these conclusions.

5. The authors show increased virus titers in MA-delta (p80) infected mice compared to MA-ancestral delta (p80). As shown, it is not obvious whether severe disease upon P80-MA-delta virus-infected mice is due to high virus titers or virus-induced inflammation or both. A side-by-side comparison of lung inflammation in MA-delta and ancestral-MA virus-infected mice/lungs is required to support these conclusions.

6. It would be interesting to know whether MA-delta and ancestral-MA viruses have differential cell tropism within the lungs and in extrapulmonary tissues. Perhaps this would also explain the basis for differential outcomes.

**Part III – Minor Issues: Editorial and Data Presentation Modifications**

Reviewer #1: Minor concerns

The introduction need to be polished. Some information, such as "experimental techniques for tracheal surgery are required for intratracheal inoculation of the AAV vector" is not accurate because AAV transfection can be accomplished well by intranasal inoculation.

Reviewer #2: 1. Introduction lacks a clear rationale. It appears that the goal of the study is to develop an MA virus that causes severe disease in 6-week-old B6 mice.

2. The rationale for using aged mice to initially adapt the viruses is not clear. While most of the advanced variants do bind to murine ACE2, the ACE2 binding ability of the ancestral variant (spike) used in this study is not provided.

3. Figure 2 and elsewhere: the authors estimate titers in BALF. Why not estimate titers in lungs?

4. Figure 4: The authors rely on lung edema and lung weight for inflammation studies. A thorough histopathological and flow cytometry evaluation of lungs/lung cells from MA-delta and ancestral-MA virus-infected mice is critical to establish lung inflammation.

5. Lines 304-317: The discussion is not overtly relevant and needs to be more aligned with the study objectives to support the results.

6. Discuss the relevance of these findings with human COVID-19 following ancestral and delta variant infections.

PLOS authors have the option to publish the peer review history of their article (what does this mean?). If published, this will include your full peer review and any attached files.

Reviewer #1: **Yes: **Jian Zheng

Reviewer #2: No
---

## [Decision Letter · Decision Letter 1]

26 Oct 2024

PPATHOGENS-D-24-01041R1TNF-α exacerbates SARS-CoV-2-infection by stimulating CXCL1 production from macrophagesPLOS Pathogens Dear Dr. Ichinohe, Thank you for submitting your manuscript to PLOS Pathogens. After careful consideration, we feel that it has merit but does not fully meet PLOS Pathogens's publication criteria as it currently stands. Therefore, we invite you to submit a revised version of the manuscript that addresses the points raised during the review process. Please submit your revised manuscript within 30 days Dec 25 2024 11:59PM. If you will need more time than this to complete your revisions, please reply to this message or contact the journal office at plospathogens@plos.org. Please include the following items when submitting your revised manuscript:*
A rebuttal letter that responds to each point raised by the editor and reviewer(s). You should upload this letter as a separate file labeled 'Response to Reviewers'. This file does not need to include responses to any formatting updates and technical items listed in the 'Journal Requirements' section below.*
A marked-up copy of your manuscript that highlights changes made to the original version. You should upload this as a separate file labeled 'Revised Manuscript with Track Changes'.*
An unmarked version of your revised paper without tracked changes. You should upload this as a separate file labeled 'Manuscript'. If you would like to make changes to your financial disclosure, competing interests statement, or data availability statement, please make these updates within the submission form at the time of resubmission. Guidelines for resubmitting your figure files are available below the reviewer comments at the end of this letter. We look forward to receiving your revised manuscript. Kind regards, Jie Sun, Ph.D.Academic EditorPLOS Pathogens Sonja BestSection EditorPLOS Pathogens Michael Malim

Editor-in-Chief

PLOS Pathogens

orcid.org/0000-0002-7699-2064 **Journal Requirements:** **Additional Editor Comments (if provided):** Please add additional discussion or "limitations of the study" section to address remaining concerns of the reviewer 2.**Reviewers' Comments:** Reviewer's Responses to Questions

**Part I - Summary**

Reviewer #1: The current revision is satisfactory and I would like to endorse its publication.

Reviewer #2: TNF exacerbates SARS-CoV-2 infection by stimulating CXCL-1 production from macrophages” by Deguchi K et al. has two primary objectives: a) to develop a reliable mouse-adapted (MA)-virus that causes severe disease in commonly used young C57BL/6 mice, and b) to investigate the basis for SARS-CoV-2-induced severe disease using the novel MA virus. Here, the authors examine underlying basis for SARS-CoV-2-induced cytokine storm and severe disease using MA SARS-CoV-2 viruses derived from different human variants. The authors show that high-passaged (p80) MA SARS-CoV-2 derived from delta variant, but not the ancestral variant, caused lethal disease in young B6 mice, while P80 virus from both backgrounds caused severe disease in BALB/c and C3H mice. In young B6 mice, the P80 MA-delta virus replicated to high titers, caused lung pathology, and triggered a robust cytokine response. The authors also showed that Myd88 and IFN-I signaling were pathogenic, early IFN-β treatment protected, and the delayed IFN-β administration caused pathology in P80 MA-delta virus-infected B6 mice. Mechanistically, the IFN-I-induced TNF-mediated CXCL-1 response was associated with severe disease in P80 MA-delta virus-infected young B6 mice

**Part II – Major Issues: Key Experiments Required for Acceptance**

Reviewer #1: No.

Reviewer #2: The authors have provided explanation for the comments raised during previous iteration. However, the authors have failed to address majority of the concerns expressed by this reviewer.

Original Comment 1: Study implications- The authors were suggested to explain the implications of the current results to human WA and Delta SARS-CoV-2 infection. Although the authors mention that they included an explanation, those modifications do not discuss implications of the current results with the outcomes in humans.

This is a missed opportunity, since highlighting the differential role of IFN-I and Myd88 in Delta MA vs WA-MA virus infected mice and providing including implications to human infections would have informed clinicians and scientist alike about differential role of these signaling pathways in different SARS-CoV-2 variant infections. The authors could have also re-written the study to highlight that severe disease upon delta variant infection could be due to differential and pathogenic role of the above mentioned pathway.

Original Comment 2: The authors were asked to provide rationale for developing a MA virus that causes severe disease in 6-week old mice as opposed to MA-virus by other labs that cause severe disease 12-20 week or older old mice. The justification of eliminating age as a factor and cellular senescence are not satisfactory, as 12-20 week mice are not old mice and they likely do not have senescent cells, unlike 20month old mice.

Original Comment 3: No explanation provided to explain why IFN-I and Myd88 may cause severe disease upon delta MA infection compared to WA-MA infection.

Original Comment 5: The authors were suggested to use TNF-/- mice or anti-TNF and anti-CXCL-1 mAb to show direct and endogenous role of these mediators in disease pathogenesis, as exogenous administration TNF, CXCL-1, and other inflammatory mediators will likely have adverse outcomes. Therefore, it is critical to block/neutralize endogenous levels of these mediators to show their clinical relevance. However, the authors could not do these studies and cite expense associated with the neutralizing antibodies as key reason to not perform the studies. These antibodies (anti-TNA) are available through BioXcell and Leinco technologies at affordable rate.

The authors instead use TAPI-2, a inhibitor of matrix matalloproteases (targets several MMPs) and TACE (targets TNF). The results obtained using TAPI-2 are not specific to TNF, and therefore the conclusions are not well justified. Moreover, TAPI-2 is given via IP route instead of IN route. Additionally, DNAse could have several off target effects.

**Part III – Minor Issues: Editorial and Data Presentation Modifications**

Reviewer #1: No.

Reviewer #2: (No Response)

PLOS authors have the option to publish the peer review history of their article (what does this mean?). If published, this will include your full peer review and any attached files.

Reviewer #1: **Yes: **Jian Zheng

Reviewer #2: No

---

## [Decision Letter · Decision Letter 2]

25 Nov 2024

Dear Dr. Ichinohe,

We are pleased to inform you that your manuscript 'TNF-α exacerbates SARS-CoV-2 infection by stimulating CXCL1 production from macrophages' has been provisionally accepted for publication in PLOS Pathogens.

Best regards,

Jie Sun, Ph.D.

Academic Editor

PLOS Pathogens

Sonja Best

Section Editor

PLOS Pathogens

Michael Malim

Editor-in-Chief

PLOS Pathogens

orcid.org/0000-0002-7699-2064

Reviewer Comments (if any, and for reference):

Reviewer's Responses to Questions

**Part I - Summary**

Reviewer #2: (No Response)

**Part II – Major Issues: Key Experiments Required for Acceptance**

Reviewer #2: (No Response)

**Part III – Minor Issues: Editorial and Data Presentation Modifications**

Reviewer #2: (No Response)

PLOS authors have the option to publish the peer review history of their article (what does this mean?). If published, this will include your full peer review and any attached files.

Reviewer #2: No

---

## [Editor Report · Acceptance letter]

3 Dec 2024

Dear Dr. Ichinohe,

We are delighted to inform you that your manuscript, "TNF-α exacerbates SARS-CoV-2 infection by stimulating CXCL1 production from macrophages," has been formally accepted for publication in PLOS Pathogens.

Best regards,

Sumita Bhaduri-McIntosh

Editor-in-Chief

PLOS Pathogens

orcid.org/0000-0003-2946-9497

Michael Malim

Editor-in-Chief

PLOS Pathogens

orcid.org/0000-0002-7699-2064